# VerifierQ: Enhancing LLM Test Time Compute with Q-Learning-based Verifiers

## Abstract

Recent test time compute approaches with verifier models have significantly enhanced the reasoning capabilities of Large Language Models (LLMs). While this kind of generator-verifier approach closely resembles the actor-critic framework in reinforcement learning (RL), the verifiers currently are used rely on supervised fine-tuning rather than on temporal difference learning. This paper introduces VerifierQ, a novel approach that integrates Offline Q-learning into LLM verifier models. We address three key challenges in applying Q-learning to LLMs: utterance-level Markov Decision Processes (MDPs), large action spaces, and overestimation bias. VerifierQ introduces a modified Bellman update, incorporates Implicit Q-learning (IQL) for efficient action space management, and integrates a novel Conservative Q-learning (CQL) formulation for balanced overestimation. Our method is among the first to apply Q-learning to LLM verifiers. This integration of RL principles into verifier models complements existing advancements in generator techniques. Experimental results on mathematical reasoning tasks demonstrate VerifierQ's superior performance compared to supervised fine-tuning approaches.

## 1 Introduction

Large Language Models (LLMs) offer a promising approach to multi-step reasoning tasks through language. However, despite their prowess in generating coherent text, LLMs face significant challenges in sustained, multi-step logical reasoning due to their underlying architecture and propensity for hallucinations (Lightman et al., 2024). Overcoming these challenges is critical for enabling the next level of agent capabilities.

One of the most important recent developments in addressing these limitations is test time compute (Snell et al., 2024; Cobbe et al., 2021). As demonstrated by OpenAI (2024), test time compute represents a new paradigm in the scaling laws of LLMs. Test time compute essentially involves using a verifier model to evaluate and select the best solutions generated by an LLM, allowing for more extensive processing and deliberation during inference. By leveraging additional computational resources at test time, LLMs can potentially perform more complex reasoning tasks with improved accuracy and reduced hallucinations.

The concept of a verifier aligns closely with recent research on multi-step reasoning, which typically employs two main components: a **generator** and a **verifier** (Lightman et al., 2024; Uesato et al., 2022; Cobbe et al., 2021). The generator produces potential solutions, while the verifier evaluates their correctness. This setup is analogous to the actor-critic framework in Reinforcement Learning (RL) (Konda & Tsitsiklis, 1999). However, unlike RL critics that use temporal-difference (TD) updates for long-term credit assignments, current verifiers in multi-step reasoning are often trained using supervised fine-tuning (SFT). This limitation might hinder the verifier's ability to guide the generator toward better long-term outcomes, particularly in complex reasoning tasks.

To address these challenges, we propose leveraging Reinforcement Learning techniques, particularly Offline Q-learning, to enhance verifier performance in long-horizon tasks. This approach draws inspiration from successful RL systems like AlphaGo Silver et al. (2016), which achieve superhuman performance by combining learning and search techniques. Recent research has focused on improving generators using methods like Monte Carlo Tree Search (MCTS) (Chen et al., 2024; Wang et al., 2024b;a; Zhang et al., 2024a). However, less attention has been given to applying RL to verifiers.

Figure 1: Illustration of State, Action (green), and Reward (orange) in a Math Problem. $+$ denotes correct (1) and $-$ denotes incorrect (0). A state generator produces an action (i.e., the next solution step). For example, $a_2$ is generated from $[P, a_1]$, and $a_3$ is generated from $[P, a_1, a_2]$. The verifier assesses the current state and action and outputs a probability of correctness.

We introduce VerifierQ, an Offline Q-learning approach that integrates classical reinforcement learning techniques with LLMs. The core research question guiding this work is: *Can Offline Q-learning improve the verifier's ability to handle multi-step reasoning tasks, and if so, how can we overcome the obstacles limiting its application to LLM value networks?*

Our work makes several key contributions: (1) We propose a flexible architecture for applying offline Q-learning on utterance-level in language models, and we resolve the large action space problem on the utterance level. (2) We present an innovative formulation of Conservative Q-learning tailored for these large action spaces, mitigating overestimation issues in offline Q-learning. (3) Our approach bridges the gap between classic critic models in reinforcement learning and verifier models in language tasks, opening new avenues for improving test-time compute in large language models.

## 2 BACKGROUND

In reinforcement learning (RL), tasks are often modeled as Markov Decision Processes (MDPs). A MDP consists of states $s \in S$, actions $a \in A$, a reward function $r(s, a)$, a transition function $P(s'|s, a)$ from state $s$ to state $s'$ with action $a$, and a discount factor $\gamma$. The goal is to find an optimal policy $\pi^*$ that maximizes the expected cumulative reward: $\pi^* = \arg\max_\pi \mathbb{E}\left[\sum_{t=0}^{\infty} \gamma^t r_t\right]$

Q-learning estimates the optimal state action value $Q$ by iteratively updating expected cumulative rewards, while the $V$ function is similar to $Q$ but just estimates from states without need of actions. Q-learning is a model-free RL algorithm commonly used to solve MDPs by minimizing the temporal difference (TD) error:

$$L_{TD}(\theta) = \mathbb{E}\left[\left(r + \gamma \max_{a'} Q(s', a'; \theta) - Q(s, a; \theta)\right)^2\right]. \tag{1}$$

Using a generator-verifier framework with large language models (LLMs) Lightman et al. (2024), we can formulate the reasoning process as an utterance-level MDP. Given a problem statement $p$, the generator produces a solution based on the problem and previous action steps, where $[p, a_1, ..., a_{i-1}]$ is the state and $a_i$ is an action. Each action is a complete sentence as shown in Figure 1. This differs from token-level approaches where actions would be individual tokens from the vocabulary, which can be a word or a subword. The state at step $i$ is the dialogue history that consists of the problem statement and all previous complete utterances generated up to that point: $s_i = [p, a_1, a_2, \ldots, a_i]$. Rewards are given at each step, with a reward of 1 for a correct step and 0 for an incorrect one. We can see the illustrated example in Figure 1, where $+$ indicates "correct" and $-$ indicates "incorrect". We will discuss advantages of utterance level in Section 4.2.

In classical RL, the critic model is trained to estimate the Q value (Konda & Tsitsiklis, 1999). In test time compute, the verifier model is trained to estimate the Q value (Snell et al., 2024). Given a problem statement, the generator produces a sequence of steps as actions. The verifier's inputs are the problem and solution steps, and it outputs correctness scores.

In general, offline Q-learning, which uses a fixed dataset to estimate Q-values, has the advantage of being more efficient to train compared to online methods since gathering new dense labeled data is difficult. However, it comes with the risk of overfitting the training data, as it lacks the exploration of new actions typically seen in online Q-learning, and it causes an overestimation problem. Recent methods involve Conservative Q Learning and Implicit Q Learning to mitigate the issue of overestimation, and this work tries to combine both approaches to overcome the overestimation problem (Kumar et al., 2020; Kostrikov et al., 2022)

## 3 RELATED WORK

Multi-step reasoning, particularly for mathematical tasks, commonly employs a generator-verifier framework to enhance LLM performance. Process Reward Modeling (PRM), which assigns rewards to each step, has been shown to outperform Object Reward Modeling (ORM), where rewards are only given for the final output (Lightman et al., 2024; Uesato et al., 2022; Cobbe et al., 2021). However, PRM requires extensive step-level labeling. Recent works mitigate this with Monte Carlo Tree Search (MCTS) for automatic labeling, improving efficiency over manual methods (Chen et al., 2024; Wang et al., 2024b;a). Nonetheless, verifiers in these approaches are trained via supervised fine-tuning (SFT), and compared to classic RL, it is similar to imitation learning.

Recent studies emphasize the role of verifiers in improving test-time compute. Cobbe et al. (2021) showed that an effective verifier can yield performance gains equivalent to increasing generator size by 30x. Similarly, Snell et al. (2024) found that optimal verifiers at test time can outperform generators 14x larger. However, these methods still rely on SFT for training verifier, limiting their effectiveness in complex, long-horizon reasoning. A more effective learning method can improve the verifier model to achieve better performance and in turn scale more efficiently.

Q-learning has seen limited use in LLMs, mainly for preference-based tasks. ILQL is the first to show implicit Q-learning can be applied to LLMs for multi-turn dialogue tasks, but the focus is on token-level actions (Snell et al., 2023). To resolve the challenges in training long-horizon reasoning due to the granularity of token-level actions, ArCHer proposed an utterance level value function, but the encoder style of value function makes estimation can compute step by step and less efficient(Zhou et al., 2024). Both works are still limited by their step-by-step computations.

In contrast, our work focuses on utterance-level actions for the verifier, and multi-reward estimation with one forward pass significantly improves training efficiency. We also integrate Implicit Q-learning (IQL) and Conservative Q-learning (CQL) to better manage large action spaces and enhance performance, offering a more scalable solution for multi-step reasoning tasks.

## 4 PROBLEM STATEMENT

The central question we need to ask is: Can Offline Q-learning improve the verifier's ability to handle multi-step reasoning tasks? If so, how can we overcome the obstacles that currently limit its application to LLM value networks?

### 4.1 OFFLINE Q-LEARNING VS IMITATION LEARNING:

The characteristic of MCTS rollout data in math problems is that it is noisy. There might be many steps that are not optimal and incorrect for solving the problem. However, stitching the optimal steps together might lead to a better solution. Those are cases that Offline Q-learning can handle better than Imitation Learning, and one of the main conclusions from Kumar et al. (2022) is that given the same noisy expert data, Offline Q-learning can outperform Imitation Learning on long horizon tasks. Intuitively, Offline RL method should learn to stitch the suboptimal paths in the noisy data to obtain a better path, and wrong answers can help offline RL what is wrong for the future. It could lead to the better performance of the verifier model even with the same amount of rollout from the generator.

### 4.2 CHALLENGES IN APPLYING OFFLINE Q-LEARNING TO LLMS:

There are several challenges in applying Offline Q-learning to LLMs. The first challenge is utterance level RL. Existing works use token level actions (Snell et al., 2023). At the token level, the action space has a smaller cardinality and is equivalent to the vocabulary size $V$, making it feasible to compute max Q-values and estimates. However, this level of granularity is too fine for long-horizon tasks (Zhou et al., 2024). On the other hand, using the utterance level allows better handling of long-term horizons, but since each utterance may contain multiple tokens, the action space grows exponentially large, making it computationally intractable (Wang et al., 2024a). Given the utterance with the number of tokens of length $n$, we will have $V^n$ actions, and a typical sentence might contain 20 tokens. This is just one utterance, but a solution contains many utterances. This creates a tension

A) Q Loss: $L_Q(\theta) = \mathbb{E}\left[\left(\frac{1}{2}(r(s,a) + \gamma V_\psi(s')) - Q_\theta(s,a)\right)^2\right]$    B) CQL Loss: $L_{CQL}(\psi) = \alpha(\mathbb{E}_{s\sim D,a\sim\mu}[L_2^{\tau_1}(Q_{\hat\theta}(s,a) - V_\psi(s))] - \mathbb{E}_{s\sim D,a\sim\hat\pi_\beta}[L_2^{\tau_2}(Q_\theta(s,a) - V_\psi(s))])$    C) Overall Components

Figure 2: Illustration of the VerifierQ architecture and modified Bellman update. Left: Bellman update, where $Q_\theta$ is updated via the TD target with $V$. Middle: CQL Loss component. The main goal is to have the lower bound $V_\psi$ as the target policy distribution while the upper bound as the data distribution. Right: Relationships among $Q_\theta$, $Q_{\hat\theta}$, and $V_\psi$. $Q_\theta$ updates through the Bellman equation as shown in (A), $V_\psi$ is updated through CQL as shown in (B), and $Q_{\hat\theta}$ is updated via soft update. The key intuition here is to leverage IQL's regression to overcome the large action space problem while CQL keeps estimation in check.

between choosing a level of granularity that is manageable and effective for long-term planning. We need to find a practical method to make the verifier model to learn on the utterance level efficiently.

The second challenge with most methods is that they rely on an actor to sample actions because it is hard to estimate the maximum Q-value when the action space is large. Each utterance level action is exponentially large as the number of tokens grows. Since datasets typically consist of rollouts with various actions, true offline learning becomes difficult. For example, for one action at $a_t$, it is needed to fix state $[P, a_1, ..., a_{t-1}]$ and sample various actions $a_t$. It is not practical to have each step have several samples while maintaining the previous steps to be the same. Given a sentence with $m$ steps, if we sample $l$ actions for each step, then we will have $l^m$ different answers for one problem if we want to apply offline learning. Additionally, finding the max Q is problematic due to the large action space, as most methods require an MCTS actor to sample the maximum value of each step (Chen et al., 2024; Wang et al., 2024a). This approach does not effectively utilize offline datasets, complicating training. Approximating $\max Q$ needs sampling, and online sampling is inefficient for training. While it is easy to roll out a complete solution, it seems to be difficult to utilize the complete rollout for training the verifier model. We need to efficiently train the verifier model with offline datasets.

The third challenge is overestimation. The overestimation problem in Q-learning is well-known, but it's particularly severe in language models. As noted in Verma et al. (2022); Zhou et al. (2024), this issue is amplified in language tasks because the Q-function is trained only on the responses in a fixed dataset, making it unlikely to predict accurate values for arbitrary strings in an utterance. In our preliminary experiments, we observed that changing just one critical token to incorrect utterances can receive a higher value than the correct ones. This overestimation becomes more pronounced at the utterance level, where the complexity and potential for incorrect value assignments are greater.

## 5 VERIFIER WITH Q-LEARNING (VERIFIERQ)

We introduce VerifierQ, a novel approach to enhancing verifier models using Offline Q-learning for Large Language Models (LLMs). Our method addresses key challenges by modifying the Q-learning algorithm and integrating it into language modeling tasks.

### 5.1 ARCHITECTURE OF VERIFIERQ

**Addressing Utterance-Level MDP:** To apply Offline Q-learning to LLMs at the utterance level, we propose a flexible architecture that integrates with language modeling tasks (Figure 2). Following Wang et al. (2024b) and Lightman et al. (2024), we utilize two tokens $+$ and $-$ to represent correct and incorrect states, with a tag token indicating estimation. The probability of the correct token out of two tokens can be interpreted as Q-values ranging from 0 to 1, aligning with the reward structure in the MCTS-generated dataset from (Wang et al., 2024b). More details are provided in the supplementary material Appendix A.1.

To address the bounded nature of outputs (0 to 1) compared to traditional Q-values, we modify the Q-learning algorithm to operate within these constraints. We propose using the $\frac{1}{2}$ of the traditional Bellman update target as the target value instead of Equation 1:

$$Q^*(s,a) = \frac{1}{2}(r(s,a) + \gamma \max_{a'} Q^*(s',a')) \tag{2}$$

where $Q^*(s,a)$ is the optimal Q-value for state $s$ and action $a$, $r(s,a)$ is the immediate reward, $\gamma$ is the discount factor, and $\max_{a'} Q^*(s',a')$ is the maximum Q-value for the next state $s'$ taking action $a'$. More details are provided in the supplementary material (Theorem 1 of Appendix A.1).

Here we need to ensure that $Q^*$ is bounded. Since $r, Q \in [0,1]$, we have $0 \leq r + \gamma \max Q \leq 2$ for $\gamma \in [0,1]$. Thus we can bound $0 \leq Q^*(s,a) = \frac{1}{2}(r(s,a) + \gamma \max_{a'} Q^*(s',a')) \leq 1$. Therefore $Q^*(s,a) \in [0,1]$, and it aligns with the model output range.

For each problem and solution sequence, we insert the tag token at the end of each step $a_i$: $[p, a_1, tag, a_2, tag, \ldots, a_n, tag]$. Then we predict the probability of $+$ or $-$ tokens as shown in Figure 1. The mean of the reward and next estimate is the target value for the Bellman update.

As shown in Figure 2, the verifier model estimates multiple Q-values for each step in the solution sequence, enabling efficient parallel computation of Q-values for multiple steps in a single forward pass. This architecture change enables VerifierQ to efficiently learn Q-values at the utterance level while maintaining compatibility with existing language modeling frameworks.

## 5.2 Algorithm of VerifierQ

**Addressing Large Action Spaces:** Traditional action selection in MDPs typically requires finding the maximum Q-value explicitly over all possible actions. In utterance-level MDPs, this leads to exponentially large action spaces of $V^n$, where $V$ is the vocabulary size and $n$ is the length of tokens in one utterance. To solve this, we employ Implicit Q-learning (IQL) (Kostrikov et al., 2022).

Our key intuition is to interpret IQL from a regression perspective. IQL approximates Q-values through regression on existing actions, mitigating the need for explicit maximum Q-value sampling and enabling efficient handling of limited per-step data. Instead of iteratively finding the maximum Q for every single action in $V^n$, it can regress the action based on the dataset and find the approximation through expectile. IQL can still approximate the maximum Q-value $\max_{a \in \mathcal{A}} Q(s,a)$ without explicitly evaluating all actions by fitting $Q(s,a)$ to the expectiles of the target values given limited data. With the vast action space, expectile regression helps interpolate and extrapolate to unobserved actions based on the observed data. This provides smooth estimates and helps handle the curse of dimensionality inherent in large action spaces. It improves sample efficiency by eliminating the need for an online algorithm to sample from. This regression-based approach makes IQL particularly well-suited for utterance-level MDPs.

We follow Snell et al. (2023) for the IQL framework to our setting, using the expectile of the Q-value to approximate the value function $V$:

$$\mathcal{L}V(\psi) = \mathbb{E}_{(s,a)\sim\mathcal{D}}\left[L_2^\tau\left(Q_\theta(s,a) - V_\psi(s)\right)\right] \tag{3}$$

where $L_2^\tau(u) = |\tau - \mathbf{1}(u < 0)|u^2$, $\tau \in (0,1)$ is the quantile level, $\mathcal{D}$ is the offline dataset, $Q_\theta$ is the learned Q-function, and $V_\psi$ is the approximated value function.

This formulation allows for efficient Q-value estimation without explicit maximization over all possible actions. Theoretically, as $\tau$ approaches 1, we have $\lim_{\tau \to 1} V_\psi(s) = \max_a Q_\theta^*(s,a)$ Kostrikov et al. (2022), ensuring that our IQL-based approach can asymptotically recover the optimal value function, even with large action spaces, given sufficient coverage in the offline dataset.

Our approach leverages regression to solve the large action space problem. By expectile regressing the reward over utterances, we can find the approximation of the maximum Q-value and minimum Q-value without needing to sample the action or iterate through all the combinations of tokens. We focus on this regression aspect, not just the data support aspect. The approximation of minimum through $\tau$ value is shown in Theorem 2 of Appendix A.1 .

**Addressing Overestimation:** Q-learning often suffers from overestimation bias, particularly severe in language models with large action spaces and limited offline datasets. To mitigate this, we incorporate Conservative Q-learning (CQL) Kumar et al. (2020) into our framework. CQL penalizes Q-values exceeding the target value, making the Q-function more conservative.

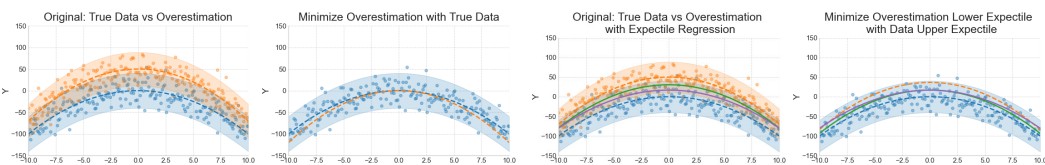

Figure 3: Illustration of our approach. Left two graphs: Orange line represents the overestimated Q-value $Q_{\hat{\theta}}$. The blue line indicates the data distribution $Q_{\theta}$. Minimizing the overestimation term brings the orange line down to the mean of data distribution. Right two graphs: The green line shows the lower expectile of the overestimated Q-value and the purple line shows the upper expectile of the data Q-value. Minimizing those two can make the orange line approach the maximum Q-value under the data distribution.

We add the following CQL term to the Bellman equation:

$$\arg\min_Q \alpha(\mathbb{E}_{s\sim D, a\sim\mu}\left[Q(s,a)\right] - \mathbb{E}_{s\sim D, a\sim\hat{\pi}_\beta}\left[Q(s,a)\right]) \tag{4}$$

Where $\mu$ is the target policy distribution and $\hat{\pi}_\beta$ is the data distribution. This term minimizes the maximum Q-value under the target policy distribution while maximizing it under the data distribution, providing a tighter bound. Unlike token-level approaches, we leverage IQL to approximate Q-values in the large action space, mitigating the need to sample a set number of actions for each state and allowing more efficient Q-value estimation for longer sequences.

Combining IQL and CQL, we propose a novel formulation that directly approximates both the lower bound Q-function and the upper bound of the data distribution using IQL:

$$L_{CQL}(\psi) = \alpha(\mathbb{E}_{s\sim D, a\sim\mu}\left[L_2^{\tau_1}\left(Q_{\hat{\theta}}(s,a) - V_\psi(s)\right)\right] - \mathbb{E}_{s\sim D, a\sim\hat{\pi}_\beta}\left[L_2^{\tau_2}\left(Q_\theta(s,a) - V_\psi(s)\right)\right]) \tag{5}$$

Here, $\tau_1$ is chosen to be close to 0 and $\tau_2$ close to 1, allowing for a more optimistic Q-value estimation within the CQL framework. This approach maintains CQL's conservatism while allowing for adaptable control through the adjustment of $\tau_1$ and $\tau_2$. The lower bound of the target policy pushes the Q-value down less aggressively, while the upper bound of the data distribution elevates it more, resulting in a more adjustable conservatism under the CQL term. For more details on the explanations of the CQL term, see Appendix 4.

Figure 3 illustrates the intuition. In the original CQL term, an overestimated Q-value would be pushed down to the data distribution. In our formulation, the lower bound of the Q-value is pushed down less aggressively, and the upper bound is elevated more, resulting in a more optimistic Q-value that approaches the maximum Q-value under the data distribution more closely. The advantage of this approach is that $\tau$ value can be adjusted to balance conservatism with optimism.

**Overall Objective:** The VerifierQ algorithm minimizes the Bellman error augmented with the CQL term. We adapt the approach of Snell et al. (2023); Kostrikov et al. (2022), using Implicit Q-learning to approximate the Q-value for each utterance level reasoning step with a separate value function while modifying the objective to incorporate the CQL term.

Like previous works, we use a separate value function $V_\psi(s')$ to approximate the Q-value $\max_{a'} Q^*(s',a')$. With our adaptation to the Bellman Update (Equation 1), the TD error is:

$$L_Q(\theta) = \mathbb{E}\left[\left(\frac{1}{2}(r(s,a) + \gamma V_\psi(s')) - Q_\theta(s,a)\right)^2\right] \tag{6}$$

We augment this with our CQL term to achieve a more conservative yet optimistic estimation with Equation 5:

$$L_{CQL}(\psi) = \alpha(\mathbb{E}_{s\sim D, a\sim\mu}\left[L_2^{\tau_1}\left(Q_{\hat{\theta}}(s,a) - V_\psi(s)\right)\right] - \mathbb{E}_{s\sim D, a\sim\hat{\pi}_\beta}\left[L_2^{\tau_2}\left(Q_\theta(s,a) - V_\psi(s)\right)\right])$$

The comprehensive objective function of VerifierQ is thus the sum of the Bellman error and the CQL term:

$$L(\theta, \psi) = L_Q(\theta) + L_{CQL}(\psi) \tag{7}$$

To enhance training stability, we employ a Polyak-averaged version of $Q_{\hat{\theta}}$ (Polyak & Juditsky, 1992). The hyperparameter $\alpha$ is set to 1 in our experiments, balancing the influence of the CQL term. The overall objective is shown in the Figure 2.

This formulation allows VerifierQ to benefit from the conservative nature of CQL while maintaining an optimistic outlook, crucial for effective Q-value estimation in large action spaces characteristic of language models. The expectile regression provides flexibility to adjust $\tau_1, \tau_2$ values as preferred. By integrating these components, VerifierQ addresses the challenges of overestimation and large action spaces in utterance-level MDPs, providing a robust framework for multi-step reasoning tasks.

# 6 EXPERIMENTS AND RESULTS

We evaluate VerifierQ on mathematical reasoning tasks from GSM8K and MATH datasets (Cobbe et al., 2021; Hendrycks et al., 2021). We compare VerifierQ with the state-of-the-art Process Reward Model (PRM) Lightman et al. (2024), using the same dataset as Wang et al. (2024b); Snell et al. (2024) for a fair comparison. We do not include Object Reward Model (ORM) since Wang et al. (2024b); Snell et al. (2024); Lightman et al. (2024) already validated PRM's effectiveness over ORM. Due to computational constraints, we use the TinyLlama-1.1B model (Zhang et al., 2024b).

## 6.1 EXPERIMENTAL SETUP

**Dataset:** We generate a test time compute set using a generator trained on MetaMath (Yu et al., 2024). The generator is finetuned on MetaMath for 2 epochs with a learning rate of 2e-5, followed by LoRA finetuning for 1 epoch to adjust the format of answer style (Hu et al., 2022). For each question in the full GSM8K test set and a 500-question subset of MATH (following Lightman et al. (2024)), we generate 256 answers. The verifier is trained on the MathShepherd dataset Wang et al. (2024b), which uses MCTS-generated data with binary rewards (1 for correct, 0 for incorrect).

**Model Architecture:** Our model consists of a Q-network and a separate value network to prevent single sample overestimation. We employ soft updates to stabilize training with rate 0.01.

**Training:** We initialize our model with MetaMath pretraining, then train with PRM on MathShepherd for 1 epoch, followed by VerifierQ training. Here are key hyperparameters. Learning rate: 2e-5 for all training phases. Batch size: 64 (crucial for Q-learning stability). Q-learning parameters: $\gamma = 0.99$, $\alpha = 1$ for the CQL term. For PRM, we continued training from 1 epoch to 2 epochs. Majority Voting uses the raw output from the generator.

**Evaluation Metrics:** We evaluate the verifier against PRM and Majority Voting using accuracy. Following Snell et al. (2024); Lightman et al. (2024), we use minimum evaluation metrics.

## 6.2 RESULTS

We evaluate VerifierQ against PRM (for epoch and two epochs) and Majority Voting on both GSM8K and MATH datasets using minimum evaluation. For VerifierQ, we use $\tau_1 = 0.3$ for GSM8K and $\tau_1 = 0.5$ for MATH.

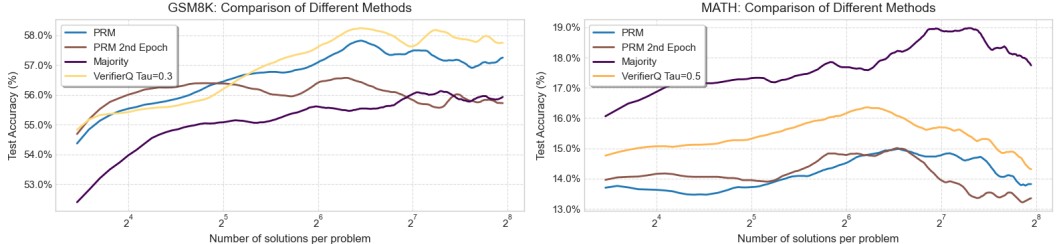

Figure 4: Comparison of different methods on GSM8K (left) and MATH (right) using minimum evaluation. Rolling average over 20 steps. For VerifierQ we use $\tau_1 = 0.3$ (left) and $\tau_1 = 0.5$ (right).

As shown in Figure 4, VerifierQ outperforms PRM (with 1 epoch), PRM 2nd Epoch and Majority Voting on both datasets. On GSM8K, VerifierQ's performance improves with an increase in the number of solutions per problem, aligning with trends observed in previous studies (Snell et al., 2024; Lightman et al., 2024; Wang et al., 2024b). While the absolute improvement margins may appear modest, a Wilcoxon signed-rank test demonstrates that VerifierQ's improvements over PRM are statistically significant (GSM8K:$W = 1500.0, p < 1.6410^{-35}$ MATH:$W = 156.5, p < 4.7010^{-43}$).

This extremely low p-value indicates that the performance differences are highly unlikely to occur by chance, suggesting that VerifierQ provides consistent, systematic improvements over baseline methods. This aligns with our theoretical expectations of improvements in verifier accuracy. We also want to point out that VerifierQ's advantage emerges as it better leverages multiple solutions for value estimation. PRM's performance gap compared to majority vote increases with the number of solutions Lightman et al. (2024); Wang et al. (2024b); Snell et al. (2024), and in our study VerifierQ's performance gap over PRM increases with the number of solutions.

We note that for MATH, all methods underperform compared to Majority Vote, possibly due to the small model size (1.1B) compared to other works. Most of the works use 7B size model as the minimum baseline, sometimes 70B. Due to practical reasons, we do not have the resources, therefor we only can work on a 1B model. Larger models are more sample efficient than smaller models Kaplan et al. (2020), and in our tasks, it means larger LLM shall learn better and have better performance given the same dataset size. They are also more capable of dealing with more complex problems that occur in MATH not in GSM8K.

In Figure 4, VerifierQ achieves the highest accuracy both on GSM8K and on MATH with different $\tau_1$ values compared to PRM (see Figure 5a in Section 7), and it also outperforms other variations (see Figure 5b in Section 7). We observe that PRM's performance decreases after the first epoch, likely due to overfitting. Therefore we will use first epoch of PRM hereafter for evaluation and ablation. Different $\tau_1$ values have different performance on two datasets (see Figure 5a in Section 7).

These results demonstrate the potential of applying classic reinforcement learning to verifier models for multi-step reasoning language tasks. They also highlight VerifierQ's effectiveness, and identify areas for future investigation, such as the impact of model size and the optimization of the values of $\tau$ for different datasets.

## 7 ABLATION STUDY

Our ablation study addresses the challenges of large action spaces, computational efficiency, and the impact of key components in VerifierQ. To be more specific, we investigate the efficiency of IQL compared to sampling approaches, the effect of the CQL term, the impact of different expectile choices, and the stability of Q-learning.

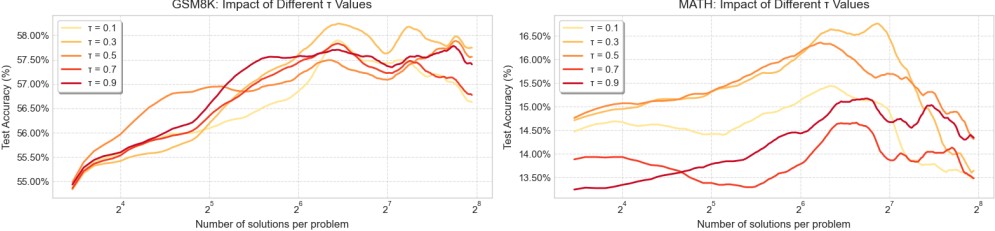

(a) Impact of different $\tau_1$ values on VerifierQ performance. Left: GSM8K dataset. Right: MATH dataset.

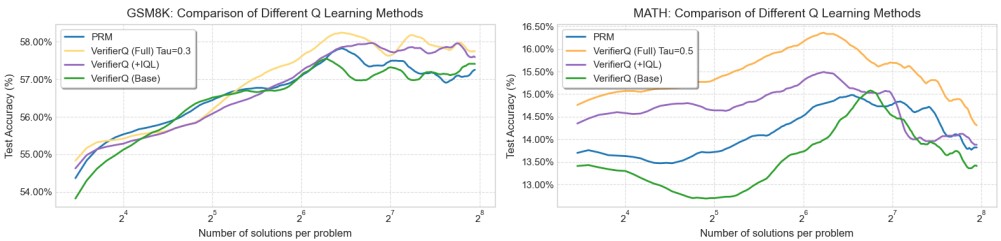

(b) Comparison of different components. PRM is blue. VerifierQ (Base) is green. VerifierQ (+IQL) is purple. VerifierQ (Full) is yellow. Left: GSM8K dataset. Right: MATH dataset.

Figure 5: Comparison of VerifierQ performance with different components

### 7.1 COMPUTATIONAL EFFICIENCY

VerifierQ's sentence-level approach offers significant computational advantages over existing utterance BERT-type and online approaches, which use [CLS] token to estimate the Q value of one

utterance Zhou et al. (2024). Traditional approaches require separate forward passes for each step's Q-value estimation, resulting in $O(n^3 m^2)$ complexity for n steps (around 6) with m tokens (around 20) each. In contrast, VerifierQ's parallel estimation through strategically placed tag tokens requires only a single forward pass, reducing complexity to $O(n^2 m^2)$. This architectural improvement enables the simultaneous computation of multiple Q-values, providing substantial efficiency gains, particularly for longer sequences. Detailed theoretical analysis and proofs comparing BERT-style, sequential decoder style, and VerifierQ are provided in Appendix C.1.

## 7.2 IMPACT OF ADJUSTABLE CQL TERM

We investigate the impact of CQL parameters $\tau_1$, which control the balance between conservatism and optimism in Q-value estimation. Our analysis focuses primarily on $\tau_1$ while fixing $\tau_2$ at 0.9, a choice guided by theoretical considerations of the CQL objective. The $\tau_2$ parameter in the CQL term controls how much we push down overestimated Q-values toward the data distribution. Since our goal is to find arg max Q, we want the upper bound of the data distribution.

Figure 5a illustrates the impact of different $\tau_1$ values on VerifierQ's performance. We examine different levels of optimism by varying $\tau_1$ (0.1, 0.3, 0.5, 0.7, 0.9) while fixing $\tau_2$ at 0.9 to tighten the lower bound of VerifierQ to the maximum of the data distribution. As shown in Figure 5a, $\tau_1 = 0.3$ generally yields better results, suggesting it approximates the maximum Q-value more effectively than other $\tau_1$ values. MATH dataset shows higher sensitivity to $\tau_1$ values, with $\tau_1 = 0.5$ performing the best. The difference in optimal $\tau_1$ values between datasets suggests that dataset-specific tuning may be necessary. It also suggests that more complex tasks (MATH) benefit from more conservative estimation.

## 7.3 COMPONENT-WISE ANALYSIS

To understand the contribution of each component, we systematically remove key elements from the full VerifierQ architecture. We conduct a comprehensive comparison of VerifierQ against other Q-learning variants to empirically validate the effectiveness of our approach, particularly the impact of the CQL term. All the rest parameters for the experiments are the same. Figure 5b shows VerifierQ (Full) outperforming both VerifierQ (Base) and VerifierQ (+IQL) on the GSM8K dataset. The following are different variations of VerifierQ:

- **VerifierQ (Base)**: Q learning only. It is a SARSA-style standard Q-learning without CQL and IQL components.

- **VerifierQ (+IQL)**: Q learning with IQL component to approximate max Q without CQL. It is an Implicit Q-learning (IQL) with $\tau = 0.9$ similar to (Snell et al., 2023) for handling the large action space problem, but without CQL since computation becomes intractable in the utterance level large action space.

- **VerifierQ (Full)**: Full VerifierQ with CQL and IQL components for tackling both the large action space and the overestimation challenges.

**Base Performance:** Standard Q-learning performs similarly to PRM on GSM8K and much worse on MATH, suggesting that naive application of Q-learning provides limited benefits. This aligns with our hypothesis that simply applying Q-learning without addressing action space and overestimation challenges is insufficient.

**Impact of IQL:** VerifierQ (+ IQL) shows notable improvements over the basic version. It is significantly higher in GSM8K but dropped to a similar level to PRM in MATH. It demonstrates that effectively handling large action spaces through IQL is crucial for language tasks, but the potential overestimation problem might make it less robust across datasets.

**Full VerifierQ:** VerifierQ (Full) shows notable and consistent improvements on both GSM8K and MATH datasets. VerifierQ outperforms other methods after $2^5$ in GSM8K and all the time in MATH. VerifierQ's superior performance demonstrates the CQL component's significant contribution and its effectiveness in reducing overestimation. The adjustable $\tau$ terms in VerifierQ allow finer control over the conservatism-optimism balance in Q-value estimation, enabling more optimistic max Q selection when appropriate for different datasets.

For the qualitative study, we added a case study and qualitative analysis in Appendix C.2. Overall we can see the progression from the basic Q-learning to the full system demonstrates how addressing each challenge (large action spaces and overestimation) leads to cumulative improvements in performance.

## 8 DISCUSSION, ETHICS, AND LIMITATIONS

VerifierQ demonstrates the potential of integrating classic reinforcement learning techniques with language models to enhance multi-step reasoning capabilities. The actor-critic model in language models could lead to more sophisticated planning and decision-making capabilities. Existing success in AI explored the actor-critic model, and actor and critic model in language models could enhance planning and decision-making capabilities, like AlphaGo.

The development and deployment of VerifierQ also raise important ethical considerations. The alignment of the reward function with human values is crucial. As the model's decision-making process becomes more complex, ensuring transparency and maintaining a human understanding what values represent becomes increasingly challenging but vastly important.

While VerifierQ demonstrates promising results, several limitations should be acknowledged: First, due to computational constraints, our experiments were limited to the TinyLlama model. Testing on larger models could potentially yield different but more likely more robust results. Second, the model's performance is highly sensitive to hyperparameter choices. We see that different $\tau_1$ choices have different performance changes in GSM8K and MATH. Resource constraints limited our ability to conduct extensive hyperparameter tuning. Finally, the fully implemented VerifierQ model is more memory-intensive and computationally expensive than the PRM method. It requires $Q_\theta, V_\psi$ to be fine-tuned while $Q_{\hat{\theta}}$ for a soft update. Compared to the SFT approach, VerfierQ needs at least 2.25 times of more VRAM for full fine-tuning due to the need to update weights, gradients, and optimizer states. Future research should focus on reducing these requirements to enhance the model's efficiency and scalability.

## 9 CONCLUSION

This work introduces VerifierQ, a novel approach integrating classical reinforcement learning techniques with language models to enhance multi-step reasoning capabilities. Our key contributions include: **(1).** A flexible architecture for applying Q-learning to utterance-level MDPs in language models. It can estimate multiple utterances level Q values with large action spaces, and easy to extend Q learning, IQL, and CQL. **(2).** An innovative formulation of Conservative Q-learning tailored for large action spaces in language tasks. It helps to reduce the overestimation in offline Q learning. **(3).** Empirical evidence demonstrating VerifierQ's effectiveness in mathematical reasoning tasks. These results highlight the potential for extending this approach to larger language models and improving test-time compute.

VerifierQ validates the integration of RL into verifier models and demonstrates its potential to enhance test-time compute results. Moreover, it bridges the gap between classic critic models in RL and verifier models in language tasks. It serves as an addition for applying RL in verifier LLMs and paves the way for actor-critic models to achieve more sophisticated artificial intelligence. As we continue to refine and expand upon this approach, VerifierQ opens up new avenues for developing more capable and robust AI systems across a wide range of complex reasoning tasks.

## REPRODUCIBILITY STATEMENT

To ensure reproducibility of our results, we provide the following details:

**Implementation Details:** The complete implementation details are available in Appendix B.1. The code and data will be made publicly available upon acceptance of this paper.

**Hardware Requirements:**

- VerifierQ experiments: Conducted on a single NVIDIA A100 GPU with 40GB memory.
- Other models (Q Learning and PRM): Can be trained on an NVIDIA RTX 4090 GPU.

**Training Time:**

- VerifierQ: Approximately 10 hours for 1 epoch.
- PRM: Approximately 4 hours for 1 epoch.

**Datasets:** We use the following publicly available datasets:

- MetaMath: `https://huggingface.co/datasets/meta-math/MetaMathQA`
- GSM8K: `https://huggingface.co/datasets/openai/gsm8k`
- MathShepherd: `https://huggingface.co/datasets/peiyi9979/Math-Shepherd`
- MATH (test subset): We use the same dataset as Lightman et al. (2024), available at `https://github.com/openai/prm800k`

**Model:** All experiments were conducted using the TinyLlama-1.1B model.

**Hyperparameters:** Key hyperparameters include:

- Learning rate: 2e-5 (constant for all training phases)
- Batch size: 64
- Discount factor ($\gamma$): 0.99
- CQL coefficient ($\alpha$): 1
- Soft update coefficient ($\alpha_{\text{soft}}$): 0.01
- IQL coefficients:
  - For GSM8K: $\tau_1 = 0.3$, $\tau_2 = 0.9$
  - For MATH: $\tau_1 = 0.5$, $\tau_2 = 0.9$

**Training Process:** The model is initialized with MetaMath pretraining, followed by 1 epoch of PRM training before VerifierQ training begins.

For any additional details or clarifications needed to reproduce our results, please refer to the code and documentation that will be made available upon acceptance.

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

# A   Appendix

## A.1   Architecture Details

To apply Offline Q-learning to LLMs at the utterance level, we propose a flexible architecture that integrates with language modeling tasks. Following Wang et al. (2024b) and Lightman et al. (2024), we utilize two tokens $+$ and $-$ to represent correct and incorrect states, with a tag token indicating estimation. The probability of the correct token can be interpreted as Q-values ranging from 0 to 1, aligning with the reward structure in the MCTS-generated dataset from Wang et al. (2024b). We compute the Q-value for each step as:

$$Q(s,a) = p(+) = \text{softmax}(\text{logit}_+) = \sigma(\text{logit}_+ - \text{logit}_-) \tag{8}$$

It is flexible to choose either softmax or sigmoid function to compute the Q-value. We use the sigmoid function in our experiments for more effficiency. The Q-value is computed for each step in the solution sequence, estimating a numerical value in the range of $(0, 1)$.

This formulation offers several advantages:

1. It allows flexible integration for Q-value estimation of utterances of arbitrary length since we can insert the step tag anywhere in the sequence.

2. It enables parallel estimation of multiple Q-values for multiple steps in a single forward pass, significantly reducing computation time.

3. This approach seamlessly integrates with existing language modeling tasks.

## A.2   Convergence of Modified Bellman Update

We first prove that our modified Bellman update converges to a fixed point.

**Theorem 1** (Convergence of Modified Bellman Update). *Let $Q^*$ be the optimal Q-function. The modified Bellman update*

$$Q^*(s,a) = \frac{1}{2}(r(s,a) + \gamma \max_{a'} Q^*(s',a')) \tag{9}$$

*converges to a unique fixed point.*

*Proof.* Let $\mathcal{T}$ be the operator defined by our modified Bellman equation:

$$\mathcal{T}Q(s,a) = \frac{1}{2}(r(s,a) + \gamma \max_{a'} Q(s',a')) \tag{10}$$

We need to show that $\mathcal{T}$ is a contraction mapping in the sup-norm $\|\cdot\|_\infty$. For any two Q-functions $Q_1$ and $Q_2$:

$$\|\mathcal{T}Q_1 - \mathcal{T}Q_2\|_\infty = \sup_{s,a} |\mathcal{T}Q_1(s,a) - \mathcal{T}Q_2(s,a)| \tag{11}$$

$$= \sup_{s,a} \left| \frac{1}{2}\gamma \left( \max_{a'} Q_1(s',a') - \max_{a'} Q_2(s',a') \right) \right| \tag{12}$$

$$\leq \frac{1}{2}\gamma \sup_{s,a} \left| \max_{a'} Q_1(s',a') - \max_{a'} Q_2(s',a') \right| \tag{13}$$

$$\leq \frac{1}{2}\gamma \sup_{s',a'} |Q_1(s',a') - Q_2(s',a')| \tag{14}$$

$$= \frac{1}{2}\gamma \|Q_1 - Q_2\|_\infty \tag{15}$$

Since $0 < \gamma < 1$, it follows that $0 < \frac{1}{2}\gamma < 1$. Therefore, $\mathcal{T}$ is a contraction mapping with contraction factor $L = \frac{1}{2}\gamma$. By the Banach fixed-point theorem, $\mathcal{T}$ has a unique fixed point, and the Q-learning algorithm will converge to this fixed point. $\square$

### A.3 OPTIMALITY OF IQL IN LARGE ACTION SPACES

Next, we prove that IQL can effectively approximate the maximum and minimum Q-value in large action spaces.

**Theorem 2** (IQL Optimality). *We can directly get the following result from the proof in (Kostrikov et al., 2022). As the quantile level $\tau$ approaches 1, the IQL value function $V_\psi$ converges to the maximum Q-value:*

$$\lim_{\tau \to 1} V_\psi(s) = \max_{a \in \mathcal{A}, \pi_\beta(a|s) > 0} Q^*(s, a) \tag{16}$$

*Additionally, as $\tau \to 0$, the IQL value function $V_\psi$ converges to the minimum Q-value:*

$$\lim_{\tau \to 0} V_\tau(s) = \min_{a \in \mathcal{A}} Q^*(s, a) \tag{17}$$

*Proof Sketch.* Following Lemma 1 of Kostrikov et al. (2022), we can show a modified Lemma. Let X be a real-valued random variable with bounded support and infimum $x^*$. Since X is bounded below and $m_\tau$ approaches the lower bound as $\tau \to 0$, we have:

$$\lim_{\tau \to 0} m_\tau = \inf\{x : F_X(x) > 0\} = x^* \tag{18}$$

For all $\tau_1$ and $\tau_2$ such that $0 < \tau_1 < \tau_2 < 1$, we can get $m_{\tau_1} \leq m_{\tau_2}$. Therefore, as $\tau \to 0$, the limit of $m_\tau$ converges to the infimum of the random variable X.

In addition, using Lemma 2 of Kostrikov et al. (2022), we can show that the IQL value function $V_\psi$ converges to the minimum Q-value as $\tau \to 0$:

**Lemma 3.** *For all s, $\tau_1$ and $\tau_2$ such that $0 < \tau_1 < \tau_2 < 1$, we have $V_{\tau_1}(s) \leq V_{\tau_2}(s)$.*

Since $Q^*(s, a)$ is bounded below, the minimum Q-value exists and is finite. Therefore, as $\tau \to 0$, the IQL value function $V_\psi$ converges to the minimum Q-value:

$$\lim_{\tau \to 0} V_\tau(s) = \inf_{a \in \text{supp}(\pi_\beta)} Q^*(s, a) \tag{19}$$

So we have:

$$\lim_{\tau \to 0} V_\tau(s) = \min_{a \in \mathcal{A}, \pi_\beta(a|s) > 0} Q^*(s, a) \tag{20}$$

$\square$

### A.4 CONSERVATIVE YET OPTIMISTIC Q-VALUES WITH MODIFIED CQL

Finally, we present a proposition about our modified CQL approach and its potential to lead to conservative yet optimistic Q-values.

**Proposition 4** (Modified CQL Bounds). *The modified CQL objective with expectile levels $\tau_1$ (close to 0) and $\tau_2$ (close to 1) aims to provide both lower and upper bounds on the true Q-function $Q^*(s, a)$:*

$$\max_{a \sim \hat{\pi}_\beta} Q_\theta(s, a) \lesssim Q^*(s, a) \lesssim \min_{a \sim \mu} Q_{\hat{\theta}}(s, a) \tag{21}$$

*where $\lesssim$ denotes "approximately less than or equal to".*

**Remark 5** (Supporting Arguments and Intuitions). The original CQL objective is:

$$\arg\min_Q \alpha(\mathbb{E}_{s \sim D, a \sim \mu}[Q(s, a)] - \mathbb{E}_{s \sim D, a \sim \hat{\pi}_\beta}[Q(s, a)]) \tag{22}$$

Where $\mu$ is the target policy distribution and $\hat{\pi}_\beta$ is the data distribution. Intuitively, this term finds the maximum Q-value under the target policy distribution $\mathbb{E}_{s \sim D, a \sim \mu}[Q(s, a)]$ and minimizes it since it is usually overestimated. To get a tighter bound, it pushes the Q-value up under the data distribution $\mathbb{E}_{s \sim D, a \sim \hat{\pi}_\beta}[Q(s, a)]$.

For large action spaces, CQL typically uses importance sampling to estimate $\mathbb{E}_{s \sim D, a \sim \mu}[Q(s, a)]$ with $\log \sum a \exp(Q(s, a))$ at every state (Kumar et al., 2020). However, unlike token-level approaches, we leverage IQL to approximate Q-values in the large action space. This mitigates the

requirement to sample a set number of actions for each state and allows for more efficient Q-value estimation for longer sequences.

We propose a novel formulation that directly approximates both the lower bound Q-function and the upper bound of the data distribution using IQL with different $\tau$ values for each term in CQL objective. The goal remains the same: finding the overestimated Q-value under the target policy to minimize it and tighten the bound with the data distribution. However we want to give control on the level of the tightening of the bound.

Our modified CQL objective is:

$$L_{CQL}(\psi) = \alpha(\mathbb{E}_{s \sim D, a \sim \mu}[L_2^{\tau_1}(Q_{\hat{\theta}}(s, a) - V_{\psi}(s))] \\ - \mathbb{E}_{s \sim D, a \sim \hat{\pi}_{\beta}}[L_2^{\tau_2}(Q_{\theta}(s, a) - V_{\psi}(s))]) \tag{23}$$

The first term, with $\tau_1$ close to 0, approximates the lower bound of $Q_{\hat{\theta}}$. It acts as an upper bound on the target policy which is typically overestimated. This suggests:

$$V_{\psi}(s) \lesssim \min_{a \sim \mu} Q_{\hat{\theta}}(s, a) \tag{24}$$

The second term, with $\tau_2$ close to 1, approximates an upper bound on $Q_{\theta}$. It acts as a lower bound on the data distribution, indicating:

$$V_{\psi}(s) \gtrsim \max_{a \sim \hat{\pi}_{\beta}} Q_{\theta}(s, a) \tag{25}$$

This approach allows for a more optimistic Q-value estimation within the CQL framework. The lower bound of the target policy $\mu$ pushes the Q-value down less aggressively, while the upper bound of the data distribution $\hat{\pi}_{\beta}$ elevates the Q-value more, resulting in a more optimistic Q-value under the CQL term. This approach maintains the benefits of CQL's conservatism while allowing for adaptable optimism through the adjustment of $\tau_1$ and $\tau_2$.

The modified CQL objective aims to minimize the difference between the lower bound of the overestimated Q-values ($Q_{\hat{\theta}}$) and the upper bound of the true Q-values ($Q_{\theta}$). Minimizing this difference may lead to a more accurate estimation of $Q^*(s, a)$. We can express this as:

$$L_{CQL}(\psi) \approx \min_{a \sim \mu} Q_{\hat{\theta}}(s, a) - \max_{a \sim \hat{\pi}_{\beta}} Q_{\theta}(s, a) \tag{26}$$

As this difference approaches zero, it suggests that the the lower bound of the overestimation of Q-values is being reduced to the extent supported by the data, and we should have $\max_{a \sim \hat{\pi}_{\beta}} Q_{\theta}(s, a) \lesssim \min_{a \sim \mu} Q_{\hat{\theta}}(s, a)$. Adjusting $\tau_1$ we could have $Q_{\hat{\theta}}(s, a)$ approximately close to the optimal maximum.

This formulation allows us to balance conservatism with optimism in Q-value estimation. The lower bound of the Q-value is pushed down less aggressively, while the upper bound is elevated more, resulting in Q-values that approach the maximum Q-value under the data distribution more closely. We can adjust $\tau_1$ and $\tau_2$ to fine-tune this balance, allowing for more adaptable Q-values under the CQL term.

It is important to note that this difference can potentially become negative. A negative value would imply that the estimated lower bound of $Q_{\hat{\theta}}$ is smaller than the estimated upper bound of $Q_{\theta}$ for some state-action pairs. While this might seem counterintuitive given the general overestimation tendency of $Q_{\hat{\theta}}$, it can occur due to the approximations introduced by the $L_2^{\tau}$ loss functions or other factors in the learning process. This suggests that the value function might be correctly valuing the in-distribution actions more highly, which is desirable, although it might introduce some pessimism in the value estimates.

This intuition provides insight into why our modified CQL approach might lead to a bit more optimistic Q-values. However, a rigorous mathematical proof would require further development and analysis.

# B APPENDIX

## B.1 ALGORITHM AND IMPLEMENTATION DETAILS

---

**Algorithm 1** VerifierQ

---

**Input:** Dataset $D$, Q-network $Q_\theta$, target Polyak-averaged Q-network $Q_{\hat{\theta}}$ with $\alpha_{\text{soft}}$, value network $V_\psi$, IQL coefficients $\tau_1$ and $\tau_2$, CQL coefficient $\alpha$

Initialize Q-network $Q_\theta$, target Q-network $Q_{\hat{\theta}}$, value network $V_\psi$

Initialize target Q-network parameters $\hat{\theta} \leftarrow \theta$

**for** each training step **do**

    Sample batch of state-action pairs $S = (s_1, s_2, s_3) \sim D$ and rewards $R = (r_1, r_2, r_3) \sim D$

    # TD Target.

    Compute target Q-values in parallel: $y = \frac{1}{2}(r + \gamma V_\psi(S'))$         $\triangleright$ Equation 2

    TD Loss: $L_Q(\theta) = \frac{1}{2}(Q_\theta(S) - y)^2$         $\triangleright$ Equation 6

    # CQL Term.

    Compute CQL $\mu$ with IQL: $L_\mu = L_2^{\tau_1}(Q_{\hat{\theta}}(S) - V_\psi(S))$         $\triangleright$ Equation 3

    Compute CQL $\hat{\pi}_\beta$ with IQL: $L_{\hat{\pi}} = L_2^{\tau_2}(Q_\theta(S) - V_\psi(S))$         $\triangleright$ Equation 3

    CQL Loss: $L_{CQL}(\psi) = \alpha(L_\mu - L_{\hat{\pi}})$         $\triangleright$ Equation 5

    # Update networks

    Update Q-network: $\theta \leftarrow \theta - \nabla_\theta L_Q(\theta)$

    Update value network: $\psi \leftarrow \psi - \nabla_\psi L_{CQL}(\psi)$

    Update target Q-network: $\hat{\theta} \leftarrow (1 - \alpha_{\text{soft}})\hat{\theta} + \alpha_{\text{soft}}\theta$

**end for**

---

As described in Section 2, the state at step $i$ is the concatenation of the problem statement and all tokens generated up to that point: $s_i = [p, a_1, a_2, \ldots, a_i]$. As illustrated in Figure 1, $s_1$ consists of $p$ and $a_1$, $s_2$ consists of $p$, $a_1$, and $a_2$, and so on. The reward $r_i$ is 1 if the token $a_i$ is correct and 0 otherwise. This approach leverages the decoder architecture's ability to generate the next token based on the previous tokens.

For the hyperparameters, we use the following settings:

- Discount factor: $\gamma = 0.99$
- CQL coefficient: $\alpha = 1$
- Soft update coefficient: $\alpha_{\text{soft}} = 0.01$
- Batch size: 64
- Optimizer: AdamW with a constant learning rate of $2e - 5$ for all training phases
- IQL coefficients:
    - For GSM8K: $\tau_1 = 0.3$, $\tau_2 = 0.9$
    - For MATH: $\tau_1 = 0.5$, $\tau_2 = 0.9$

We initialize the model with MetaMath pretraining and train it with PRM for 1 epoch before starting VerifierQ training. All experiments are conducted using the TinyLlama-1.1B model.

# C APPENDIX

## C.1 COMPUTATIONAL EFFICIENCY ANALYSIS

VerifierQ's architecture offers significant computational advantages over both BERT-style encoder and traditional sequential decoder approaches.

For a solution sequence with $n$ steps, where each step contains $m$ tokens on average, let $C(m)$ denote the computational cost of a forward pass through m tokens. The self attention in transformer in general has Big-O of $O(L^2d + Ld^2)$ where as L is the sequence Length and d is depth. Since we have depth to be the same and varies of sequence length, we can view it in our case as $O(L^2)$. The traditional sequential approach computes:

$$Q_{\text{seq}}(s_i, a_i) = f_\theta(\text{concat}(s_i, a_i)) \tag{27}$$

where $f_\theta$ represents the model's forward pass.

For example, consider a three-step solution:

*Problem: A robe takes 2 bolts of blue fiber and half that much white fiber. How many bolts in total does it take?*

*Step 1: It takes $2/2 =<< 2/2 = 1 >> 1$ bolt of white fiber $< tag > +$*

*Step 2: So the total amount of fabric is $2 + 1 =<< 2 + 1 = 3 >> 3$ bolts of fabric $< tag > +$*

*Step 3: The Answer is: $3 < tag > +$*

**BERT-style Encoder Approach:** For encoder architectures like BERT, Q-values are typically estimated through a [CLS] token. As shown in Zhou et al. (2024), they are using RoBERTa based model to estimate the utterance level with "[CLS]" token:

$$Q_{\text{encoder}}(s_i, a_i) = f_\theta([\text{CLS}] + \text{concat}(s_i, a_i)) \tag{28}$$

where the [CLS] token representation is used for value estimation.

As an example would estimate like this:

- Pass 1: $Q(s_1, a_1) = f_\theta([\text{CLS}] + [p, a_1])$
- Pass 2: $Q(s_2, a_2) = f_\theta([\text{CLS}] + [p, a_1, a_2])$
- Pass 3: $Q(s_3, a_3) = f_\theta([\text{CLS}] + [p, a_1, a_2, a_3])$

This requires separate encoding for each step since we can have only one [CLS] each time. Since we have $n$ steps, each step contains $m$ tokens on average, and $C(m) = O(L^2)$, we can have following:

$$\text{Cost}_{\text{encoder}} = \sum_{i=1}^{n} C(im) = \sum_{i=1}^{n} (im)^2 = m^2 \sum_{i=1}^{n} i^2 = m^2 \frac{n(n+1)(2n+1)}{6} = O(m^2 n^3) \tag{29}$$

**Sequential Decoder Approach:** Traditional decoder architectures estimate Q-values at sequence endpoints by predicting the last embedding:

$$Q_{\text{seq}}(s_i, a_i) = f_\theta(\text{concat}(s_i, a_i)) \tag{30}$$

An example would be like this:

**Sequential Decoder:**

- Pass 1: $Q(s_1, a_1) = f_\theta([p, a_1])$
- Pass 2: $Q(s_2, a_2) = f_\theta([p, a_1, a_2])$
- Pass 3: $Q(s_3, a_3) = f_\theta([p, a_1, a_2, a_3])$

Where as the Q value estimation is from the linear head of the last token embeddings.

Similarly requiring n separate computations:

$$\text{Cost}_{\text{seq}} = \sum_{i=1}^{n} C(im) = \sum_{i=1}^{n} (im)^2 = O(m^2 n^3) \tag{31}$$

**VerifierQ's Parallel Approach:** Following Wang et al. (2024b), VerifierQ uses decoder architecture with strategically placed tag tokens enabling parallel estimation.

$$[Q_{\text{parallel}}(s_1, a_1), ..., Q_{\text{parallel}}(s_n, a_n)] = f_\theta(\text{concat}(s_1, \text{tag}, a_1, ..., s_n, \text{tag}, a_n)) \tag{32}$$

As an example it would be like following: **VerifierQ:**

$$[Q(s_1, a_1), Q(s_2, a_2), Q(s_3, a_3)] = f_\theta([p, a_1, \text{tag}, a_2, \text{tag}, a_3, \text{tag}]) \tag{33}$$

It would require only a single forward pass:

$$\text{Cost}_{\text{parallel}} = C(nm) = O(n^2 m^2) \tag{34}$$

This parallelization provides several advantages:

- Reduced complexity: From $O(n^3 m^2)$ to $O(n^2 m^2)$
- Parallel computation: Simultaneous Q-value estimation for all steps
- Decoder architecture benefits: Natural alignment with autoregressive generation

To quantify these benefits, consider our preliminary experiments using the MathShepherd dataset (Wang et al., 2024b), where solutions average 6.2 steps per problem. The sequential approach requires:

- Two forward passes per step (current Q and next Q) for Q-learning
- Total passes per problem = 6.2 steps × 2 passes = 12.4

In contrast, VerifierQ computes all Q-values in a single forward pass. This theoretical reduction from approximately 12 passes to 1 aligns with our preliminary observations of approximately 10× reduction in training time. This efficiency gain would become even more pronounced for longer solution sequences where n is large, demonstrating the scalability of our approach.

## C.2 QUALITATIVE OVERESTIMATION ANALYSIS

We conduct a qualitative study on overestimation between PRM and VerifierQ by replacing important tokens in the solution sequence with incorrect ones. Figure 6 reveals that PRM generally assigns higher Q-values to incorrect tokens, while VerifierQ assigns lower values.

This analysis compares three configurations: 1. PRM (baseline) 2. VerifierQ without CQL (ablation) 3.VerifierQ (full model). We want to see how well the correct value is differentiated from other incorrect values, so we use $\Delta = Correct - mean(Incorrect)$ to measure the difference between the correct answer and the mean of the incorrect answers. Then we want to see how much this value compared to mean to see whether it is a large differentiation or not, and we use Percentage as $\Delta/mean$

Our analysis reveals several key patterns:

- **PRM**: Assigns relatively high Q-values (mean: 0.24) with average discrimination between correct and incorrect tokens ($\Delta = 0.03$). Percentage is 11.9%.

- **VerifierQ without CQL**: Shows higher overall Q-values (mean: 0.39) but slightly better discrimination ($\Delta = 0.05$). Percentage 12.6%.

- **VerifierQ with CQL**: Demonstrates: 1. Lower overall Q-values (mean: 0.20) 2. Stronger discrimination between correct and incorrect tokens ($\Delta = 0.06$) 3. More aggressive penalization of incorrect tokens (Percentage: 30.6%)

The CQL term's impact is twofold:

1. **General Conservative Estimation**: Reduces overall Q-values to mitigate general overestimation. We can see the without CQL term the Q learning has overestimated significantly, and CQL term brought down overestimation. However adding CQL the correct value is about the same as PRM but incorrect ones are lower. It penalizes the incorrect tokens more.

2. **Enhanced Discrimination**: Increases the gap between correct and incorrect token estimates. The Percentage gap of correct tokens has almost doubled compared to the PRM and VerifierQ without CQL.

This analysis suggests that while CQL does lead to generally lower Q-values, its primary benefit is the enhanced ability to distinguish between correct and incorrect solutions. The increased gap between correct and incorrect Q-values (Percentage = 30.6% compared to 11.9% and 12.6%) demonstrates that CQL improves the model's discriminative capability while maintaining reasonable estimates for valid solutions.

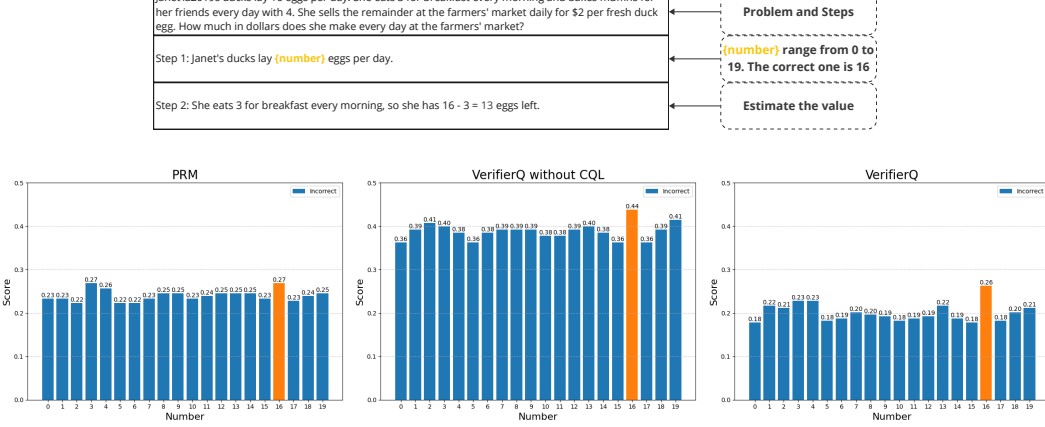

Figure 6: Overestimation case study: PRM (left) vs VerifierQ without CQL (middle) vs VerifierQ (right). Orange indicates correct value, blue indicates incorrect value.

