# OpenReview forum: "VerifierQ: Enhancing LLM Test Time Compute with Q-Learning-based Verifiers"
_ICLR.cc/2025/Conference — Submitted to ICLR 2025_

### Official Review · Reviewer_D9F5 · 2024-10-29

**Soundness:** 2
**Presentation:** 2
**Contribution:** 3
**Rating:** 3
**Confidence:** 4

**Summary:**

This paper proposes the VerifierQ, which use offline Q-learning to train the Verifiers of LLMs. The authors address three key challenges in applying Q-learning to LLMs: (1) handling utterance-level Markov Decision Processes (MDPs), (2) managing large action spaces, and (3) mitigating overestimation bias. The experiments on GSM8K and MATH show that the methods are better than baselines.

**Strengths:**

1. Combining RL and LLM is an important topic,  and VerifierQ validates the integration of RL into verifier models and demonstrates its potential to enhance test-time compute results.

2. The authors address three practical challenges for the combinition of RL and LLM. Some of the design seems to be reasonable (Some Theorem shown that the methods can converge under mild conditions)

**Weaknesses:**

1. The paper writing needs to be improved. For example, line-068: key contributions:(1) We propose  ==> key contributions: (1) We propose; line-183: doesn't ==> does not; line-188: it's ==> it is; line-191: we've ==> we have; line-398: tau = 0.5 ==> tau_1 = 0.5; etc...

2. Some important details for the methods are missing. For example, Section 5.1 is hard to understand: (1) the authors say "We propose using the mean of the current Q-value and the traditional Bellman update as the target value" in line-212, and "The mean of the reward and next estimate serves as the target value for the Bellman update." in line-240. (2) The Figure 2 demonstrates the architecure, but it is not clear and not consistent with the text writing.

3. Why directly applying IQL and CQL can address the Large Action Spaces and overestimation? It is know that large action space is very hard for RL. IQL and CQL are generally applied to traditional RL setting, with a few actions in the action space.

4. The experiments are weak. Only testing the methods on two datasets with small LLMs.

**Questions:**

see the above.

---

> ### Author Response · Authors · 2024-11-22
> **Partial Response 1**
>
> # Response to Reviewer Comments
>
> Thank you very much for taking your time and detailed review! We appreciate your recognition of the importance of integrating reinforcement learning (RL) with large language models (LLMs) and your acknowledgment of our efforts to address key challenges in this area. We also believe a more classic RL approach to improve generator verifier has promising potential. We address your concerns point by point below.
>
> ## Writing and Presentation
> We apologize for the issues in writing and presentation that you pointed out. We have thoroughly revised the manuscript to improve the writing quality, fixed grammatical errors, and ensured consistency throughout the paper.
>
> ## Technical Clarity
>
> ### Section 5.1
> - Thank you for pointing out. We understand that Section 5.1 may have been difficult to follow, and we revised it to enhance clarity and provide more detailed explanations.
> - We have clarified that we use the mean of the immediate reward and the discounted next estimate as the target value in Equation 2.
> - This modification helps bound the Q-values within the desired range of [0, 1].
> - We have added a mathematical justification in Section 5.1.
>
> ### Figure 2 Improvements
> We acknowledge that Figure 2 may not have been clear. We have redesigned Figure 2 to more accurately represent the architecture and flow of computations:
> - Part A: Illustrates the Q-loss, which uses the Bellman update
> - Part B: Depicts the CQL loss, which updates the value function  $V_\psi$​.​
> - Part C: Shows the overall interaction between components, with clear labels indicating where Parts A and B occur within the framework
> - Added detailed annotations and descriptions to guide readers through the figure
>
> ## IQL and CQL Question
>
> ### Large Action Spaces:
> 1. **Use of IQL:**
>   In language models, the action space (possible utterances) is extremely large, making traditional RL methods challenging. IQL addresses this by avoiding explicit maximization over actions, which is computationally infeasible in large action spaces.
>
> 2. **Regression Perspective:**
>   We interpret IQL from a regression perspective, focusing on its ability to approximate Q-values across the entire action space through expectile regression. Since observed actions are sparse samples of the vast action space, expectile regression helps interpolate and extrapolate to unobserved actions based on patterns in the observed data. This provides smooth estimates and helps handle the curse of dimensionality inherent in large action spaces. We discuss this part more in Section 5.2 Addressing Large Action Spaces part. We added a few intuition explanations to make it more explicit.
>
>
> 3. **Practical Implications:**
>  In our case, the discrete action space is vast due to numerous possible token combinations at each step. For example, the MathShepherd dataset contains about 15 solutions per problem, but each step can vary significantly, resulting in many unobserved actions.IQL enables us to estimate Q-values without exhaustively exploring all possible actions.
>
>
>
> ### Overestimation:
> **Use of CQL:**
> We adapt CQL to mitigate overestimation bias, which is especially problematic in large action spaces. CQL penalizes overestimated Q-values for out-of-distribution actions by encouraging the model to prefer actions within the data distribution. The original CQL paper demonstrates its effectiveness in continuous action spaces, which are analogous to large discrete action spaces in terms of complexity. CQL is not limited to settings with few actions; it is designed to handle scenarios where the action space is large or continuous.
>
>
> Our approach leverages IQL and CQL to effectively manage large action spaces and mitigate overestimation bias in LLMs. By combining these methods, we address the challenges posed by the vast action space in language models. Our empirical results demonstrate that this method outperforms baselines, indicating its effectiveness.

---

> > ### Author Response · Authors · 2024-11-22
> > **Partial Response 2**
> >
> > ## Experimental Validation and Scope
> >
> > ### Datasets Used:
> > We selected GSM8K and MATH as they are widely recognized benchmarks:
> > - [1] uses MATH
> > - [2] uses GSM8K and MATH
> > - [3] uses GSM8K, MATH, and MBPP
> > - [4] uses MATH
> > - [5] uses GSM8K and MATH
> > - [6] uses GSM8K, MATH, GK2023, and OCW
> >
> > GSM8K and MATH are most consistently used, which guided our focus.
> >
> > ### Model Size and Resource Constraints:
> > Due to computational resource limitations, we conducted experiments using models with approximately 1 billion parameters. We agree that experimenting with larger models could further demonstrate the effectiveness of VerifierQ. However, not all research groups have access to the substantial computational resources and GPUs required for training larger models.
> >
> >
> > ### Significance of Results:
> > Despite using smaller models, our experiments show that VerifierQ outperforms traditional supervised fine-tuning approaches. We have updated Section 6.2 to include statistical significance tests to strengthen our claims. The observed improvements, although obtained with smaller models, indicate the potential of our approach.
> >
> >
> >
> > ### Addressing Limitations:
> > 1. **Limitations Acknowledgment:**
> > We have explicitly acknowledged the limitations of our experimental setup in the revised paper.
> >
> >
> > 2. **Future Work:**
> > We plan to conduct experiments with larger models and additional datasets as resources become available. We believe that the trends observed with smaller models will generalize to larger ones, and we are eager to explore this in future work.
> >
> > ## Overall
> >
> > We are grateful for your feedback, which has highlighted areas where we can improve our paper. We have addressed your concerns regarding writing quality, methodological clarity, the applicability of IQL and CQL, and experimental validation.
> >
> > We hope that our responses demonstrate the value of our contributions to advancing the integration of RL techniques into LLMs for enhanced reasoning capabilities.
> >
> > ## References
> >
> > [1] Let's Verify Step by Step https://openreview.net/forum?id=v8L0pN6EOi, Jan 2024
> > [2] Math-Shepherd: Verify and Reinforce LLMs Step-by-step without Human Annotations https://aclanthology.org/2024.acl-long.510/ Feb 2024
> > [3] Q*: Improving Multi-step Reasoning for LLMs with Deliberative Planning https://arxiv.org/abs/2406.14283, Jul 2024
> > [4] Scaling LLM Test-Time Compute Optimally can be More Effective than Scaling Model Parameters https://arxiv.org/abs/2408.03314, Aug 2024
> > [5] Metamath: Bootstrap your own mathematical questions for large language models https://openreview.net/forum?id=N8N0hgNDRt  Jan 2024
> > [6] Alphamath almost zero: process supervision without process https://openreview.net/forum?id=VaXnxQ3UKo , Sep 2024

---

### Official Review · Reviewer_ap5k · 2024-11-02

**Soundness:** 2
**Presentation:** 1
**Contribution:** 3
**Rating:** 5
**Confidence:** 3

**Summary:**

VerifierQ introduces a new approach for enhancing verifier models in Large Language Models (LLMs) through Offline Q-learning, focusing on utterance-level tasks, featuring three key contributions:
- Adaptation of Q-learning to work on utterance-level for Language Modeling
- Handling Large Action Space with Implicit Q-Learning (IQL): To handle large action spaces in LLMs VerifierQ integrates Implicit Q-Learning (IQL), which approximates Q-values without explicit maximization, making Q-value estimation feasible even with extensive token vocabularies.
- Conservative Q-learning for Overestimation Control: VerifierQ incorporates a Conservative Q-learning (CQL) term to counter overestimation bias, making Q-value updates more conservative. In contrast to classic CQL, VerifierQ add a term to enabling balance between conservatism and optimism

**Strengths:**

The combined use of Implicit Q-learning (IQL) and Conservative Q-learning (CQL) is innovative and could offer promising results for balancing efficiency and accuracy in Q-value estimation for language models.

**Weaknesses:**

The paper is difficult to follow and some concepts and sections lack clear explanations

There are missing citations, often important concepts are introduced without a proper citation, for examples:
-- line 094: generator-verifier framework
-- line 342: Polyak-average

Incomplete References: Published versions of cited works are not referenced.
Line 107: the two articles you are citing have been published respectively to NIPS 2020 and NIPS 2021, but you do not report this in the references.

Figures are hard to read, with text and images being too small (especially Figures 3 and 5).

The ablation section is inadequately detailed and lacks clarity.
Experiments described in figure 5 investigate parameter tuning rather than a true ablation, as they do not remove components to measure their impact, but rather modify \tau_1, without exploring the effect of \tau_2
The second part is unclear; it does not specify which components were from the original architecture, nor does it provide sufficient comments on results
Authors should enrich the ablation section by clearly detailing removed components, explaining the experimental setup, and discussing the results.

Detailed comments on the text (numbers refer to lines in the pdf):

044: I could not understand to which work "This approach" refers, please clarify.
068: Should this be "offline Q-learning" instead of "Q-learning"?
097: Please provide a clear definition for the input and output of the verifier
103: When you talk about offline Q-Learning, are you referring to a particular work or generally about offline techniques built upon Q-learning? Please clarify.
104: "More efficient" in what terms? Please specify.
Figure 1:
    What is the source of action a_2 ?
    Why does a state generator generate an action?
    Why is the generator output described as a single action a_1, when for example at line 094 you say that "the generator's output can be treated as a SEQUENCE of actions" ?
164: Could you clarify or highlight where "utterance-level RL" is defined?
175-178: This concept is unclear; can you explain it further?
199: You mention modifying the offline Q-learning algorithm without defining offline Q-learning itself.
236: "taking action a'" is missing after ".. the next state s'"
325: When you talk about step, please refresh to what you are referring
363: What’s the rationale for splitting the fine-tuning process (first full, then LoRA)?
435: This is the first mention of “BERT-type” approaches. Please add context and references, and include utterance-level online approaches in related work if you plan to compare against them here.
440: Can you substantiate the claim of saving "10x training time"? What is the comparison baseline?

**Questions:**

Q1) Distributional shift is a typical issue for offline RL, I guess this is an issue for this apprach as well, please clarify.

Q2) Ablation: i) Did you explore changes to /tau_2? ii) How challenging is it to fine-tune /tau? iii) One dataset showed substantial differences with incorrect /tau_1 values. Have you tested other datasets in order to prove how sensitive is your model to /tau selection?

Q3) Possible limitations of the approach
    - How much is the model sensible to parameters changes?
    - How does the method compares to other approaches in terms of memory rquirements ?

Q4) Would it be possible to test the code for reproducibility ?

---

> ### Author Response · Authors · 2024-11-22
> **Partial Response 1**
>
> # Response to Reviewer Comments
>
> We sincerely thank you for your detailed review and for recognizing the innovative aspects of our work, particularly the combined use of Implicit Q-Learning (IQL) and Conservative Q-Learning (CQL) to enhance verifier models in Large Language Models (LLMs). Your constructive feedback is greatly appreciated and has helped us identify areas for improvement!
>
> ## Weakness Addressed
>
> ### 1. Paper Clarity and Explanations
> We apologize for the lack of clarity in some sections of the paper. We are committed to revising the manuscript to enhance readability and ensure that all concepts are clearly explained. Specifically we focus on improving clarity. We added more detailed explanations and examples where necessary.
>
> ### 2. Missing Citations and References
> - Thank you for bringing this to our attention. We apologize for the oversight and have added the relevant citations to provide proper context.
> - We have updated our references to include the published versions of these works.
>
> ### 3. Figure Readability
> We acknowledge that the figures were difficult to read. We have revised all figures to enhance their readability. We increased font size and some figure sizes, in addition, we added more context to improve the readability.
>
> ### 4. Ablation Study Improvements
> Thank you for this valuable feedback. We have substantially revised the ablation study section 7.2:
>
> - For the clarification of Ablation components, we have clearly specified which components are being ablated in each experiment. The components "Q-learning" and "IQL" have been renamed to more descriptive terms to indicate what is being removed or modified.
>
> - For the exploration of $\tau_2$ we added more theoretical explanations in section 7.2. The reason why we fix $\tau_2$ is because of the CQL part. The effect of $\tau_2$ term in CQL is to push down the overestimation, and we want it to push down toward the maximum of the data distribution since we are looking for $\arg{max} Q$. Fixing $\tau_2$ to a high value can ensure it would not push further down from the maximum data distribution. We had a very preliminary study before this paper, much smaller parameters were trained, and lower $\tau_2$ did not perform as good as higher values. That’s why we decided not to pursue different $\tau_2$ based on theoretical analysis and preliminary experiments.
>
> - The experimental setup is now thoroughly explained, detailing how components were removed or altered. We have provided comprehensive discussions of the results, highlighting the impact of each component on performance.
>
> ## Detailed Text Revisions
> We really appreciate your detailed comments on the text! We modified most of your suggestions directly on the manuscript for most of the points. Line number refers to original lines in the pdf, and comments in parentheses ‘()’ are responses and line numbers of updated PDF. Changes include:
> - 044 (changed)
> - 068 (changed)
> - 097 (added on line 101)
> - 103 (changed)
> - 104 (changed)
> - Figure 1 (added more description)
> - 164 (added in Section 2, 88-97)
> - 175-178 (added example)
> - 199 (defined in background section 102-107)
> - 236 (changed)
> - 325 (changed)
> - 363 (changed to 342, LoRA is primarily for adjusting answer format)
> - 435 (changed at 431, added additional analysis in Appendix C.1)
> - 440 (added detailed analysis in Appendix C.1)
>
> ## Additional Clarifications
>
> ### Action Generation Process
> The source of action $a_2$ is produced from Problem P and action a₁. Autoregressive model uses previously generated text to produce the next text. And action $a_3$ is the output of the generator from concating $P, a_1, a_2$ as input.
>
> We added more descriptions on:
> - How each actions are produced
> - Utterance level MDP in section 2
> - Analysis on BERT style and other approaches in Appendix C.1
> - Training time analysis in Appendix C.1

---

> > ### Author Response · Authors · 2024-11-22
> > **Partial Response 2**
> >
> > ## Responses to Questions
> >
> > ### Distributional Shift
> > We agree. In our approach we try to mitigate it through CQL and IQL. CQL penalizes out-of-distribution actions and encourages the model to focus on the action observed in the dataset. IQL part also reduces overfitting through the expectile regression since it avoids out-of-sample actions. From our comparison of Q learning method quantitatively and qualitatively, it seems that distribution causes overestimation and it hurts performance for Q learning a lot, especially due to large action space. Our approach has better performances quantitatively and qualitatively compared to Q learning -- more correct answers and lower estimated values. Quantitatively please see Figure 5, qualitatively please see the added section of Appendix C.2.
> >
> > ### $\tau_2$  Parameter Settings
> > We discuss the $\tau_2$ in the above response. Fine tuning $\tau_2$ values needs careful consideration, as it decides the lower bound of data distribution and we want the maximum lower bound based on our theoretical side. Empirically we suggest to keep $\tau_1$ small from 0.1-0.5 while keeping $\tau_2$ large from 0.9. Fine tuning $\tau$ isn’t difficult, but because it is a set parameter during training so it needs to be preset. We observed that the optimal $\tau$ values can vary between datasets, as seen with GSM8K and MATH. Different data distributions may affect how $\tau$ values influence learning. We have added discussions and experimental results in the paper to illustrate the model's sensitivity to $\tau$ across datasets.
> >
> >
> > ### Limitations Discussion
> > We have expanded the limitations section to include:
> >
> > **Parameter Sensitivity:**
> > We have addressed some limitations in Section 8; however, we appreciate your feedback and will provide a more detailed discussion. The model's performance varies with the parameter $\tau_1$ on the GSM8K and MATH datasets. We hypothesize that this discrepancy arises from differences in the data distributions and, consequently, the approximation of maximum values. This observation motivates further investigation into the relationship between data characteristics and the optimal choice of $\tau_1$.
> >
> > **Memory Requirements:**
> > Regarding memory requirements, we will include a more comprehensive analysis in the revised manuscript. Our preliminary assessment suggests that our method requires approximately 2.25 times the memory of the PRM method. Specifically, for VRAM, we need to load two models and perform soft updates on one. Therefore, for each GB of model parameters, optimizer states, and gradients, we estimate a requirement of (1GB * 2) * 4 + 1GB = 9GB. In contrast, the PRM method requires roughly 1GB * 4 = 4GB. The ratio of these requirements is 9/4 = 2.25. Consequently, our model would necessitate approximately 2.25 times more VRAM. We have added this conclusion to the section discussing limitations.
> >
> > ### Code Availability
> > We will make our code and scripts publicly available upon publication of the paper, including documentation and instructions for running experiments.
> >
> > ## Overall
> >
> > We thank you again for your valuable feedback. Your insights have helped us identify areas where we can improve the clarity and completeness of our paper. We will incorporate all the suggested changes and ensure that our revisions address your concerns. We believe that these enhancements will strengthen our work and make it more accessible to the research community.
> >
> > We hope that our responses have satisfactorily addressed your comments and that you will consider our paper favorably.

---

### Official Review · Reviewer_3LxU · 2024-11-03

**Soundness:** 3
**Presentation:** 2
**Contribution:** 3
**Rating:** 8
**Confidence:** 3

**Summary:**

The paper introduces VerifierQ, a novel method that integrates Offline Q-learning into verifier models for LLMs, significantly enhancing their multi-step reasoning capabilities. It addresses key challenges such as handling utterance-level MDPs, managing large action spaces, and mitigating overestimation bias through a modified Bellman update and a tailored formulation of Conservative Q-learning. Experimental results demonstrate that VerifierQ outperforms traditional supervised fine-tuning approaches in mathematical reasoning tasks, suggesting a promising direction for improving test-time compute in LLMs.

**Strengths:**

1- The paper is well-structured and easy to follow.

2-  The paper cleverly handles multiple challenges (utterance level RL - large action space and overestimatrion ) and proposes a novel loss function formulation that integrates both IQL and CQL, allowing for a more optimistic Q-values estimation within the CQL framework.

3- Comprehensive evaluation and ablation study.

**Weaknesses:**

Overall, I think this is a good paper but I do have the below comments:

1- Clarification on Q-learning Update Bounds:
- In page 4, Section 5.1, it is not clear to me how multiplying the q-learning update by 1/2 bounds the Q-value between (0,1) as the authors claim. Can you please explain more about the intuition behind this? A brief mathematical derivation or graphical representation could enhance understanding.

2- Clarity of the figures:
- The figures throughout the paper are difficult to read due to the small font size.  Please increase the font size. Additionally, Figure 2 A) could benefit from more context or annotations to aid in comprehension.

- Lastly, in Figures 4 and 5,  the y-axis is labeled 'Test Accuracy %', but the values (e.g., 0.53, 0.54) are presented as decimal fractions rather than percentages. The authors should clarify whether this is a typo or if the values are indeed less than 1.

**Questions:**

Please see the questions above.

---

> ### Author Response · Authors · 2024-11-22
>
> # Response to Reviewer Comments
>
> We sincerely thank you for your positive evaluation of our paper and for recognizing the strengths of our work, including its structure, the handling of multiple challenges, and the novel integration of Implicit Q-Learning (IQL) and Conservative Q-Learning (CQL) in our loss function formulation. We appreciate your thoughtful comments and suggestions, which will help us improve the clarity and quality of our paper.
>
> ## 1. Clarification on Q-learning Update Bounds
>
> The key here is the summation of each step of the Q value would be in the range of [0, 2] due to the probability output. $r \in [0,1]$, and $Q \in [0, 1]$. Technically it is bounded by $0 \leq r+gamma\*Q \leq 1+ gamma\*1 \leq 2$ given $gamma \leq 1$. And times $\frac{1}{2}$ we will have $0 \leq \frac{1}{2} (r+gamma\*Q) \leq  \frac{1}{2} (1+ gamma\*1) \leq 1$. We include this mathematical derivation in the revised manuscript Section 5.1 (Lines 224-227) to enhance clarity.
>
>
> ## 2. Clarity of Figures:
>
> - Thank you for bringing this to our attention. We have updated all the figures in the paper to increase the font sizes, ensuring they are easily readable.
> - We revised Figure 2 for more clarity. We added corresponding loss components so it is more clear. We added more context and changed the font size for Figure 2. Also you correctly point out the percentage, it should be in percentage instead of the decimals. We corrected it. Sorry for the confusion.
>
> # Overall
> We appreciate your constructive feedback and are committed to improving our paper based on your suggestions. Your insights have been invaluable in enhancing the clarity and quality of our work.
>
> Thank you again for your positive evaluation and for helping us strengthen our paper.

---

### Official Review · Reviewer_M9KV · 2024-11-11

**Soundness:** 2
**Presentation:** 2
**Contribution:** 2
**Rating:** 5
**Confidence:** 3

**Summary:**

This paper focused on integrating reinforcement learning techniques with language models to enhance multi-step reasoning capabilities. To do that, the approach of VerifierQ was proposed by formulating the problem into a utterance-level MDP and solving it with IQL and CQL. Finally, experiments were conducted on two datasets of GSM8K and MATH, comparing VerifierQ with PRM and Majority Voting.

**Strengths:**

- This is a timely important topic that the paper aims to leverage verifier models with reinforcement learning to enhance the reasoning capabilities of large language models. It's interesting to pay attention to the verifier component instead of the generator component.
- Novelty: although this work is built from the previous work on multi-step reasoning with generator-verifier, it's novel to formulate the problem into a utterance level MDP, from the perspective of verifier models, and solve it with reinforcement learning techniques.

**Weaknesses:**

- The presentation of the paper should be improved. For example, it can be more helpful if Figure 1 could include IQL and CQL and show how different modules/components interacct with each other and what's the forward pass and backward pass. The paper mentioned that the utterance level RL allows better handling of long-term horizons, so, what's the MDP definition in the utterance level? what's the difference between their utterance level and token level in this paper?
- The current results are insufficient to support the paper's claim. For example, I didn't find the results used for showing the computational efficiency of VerifierQ and others. In Figure 4, particularly on GSM8K dataset, VerifierQ performed worse than PRM when the number solutions per problem was lass then 2^5, and it became the best when the number was larger than 2^5. However, the improvement looks very marginal. On MATH dataset, VerifierQ performed worse than Majority. Eventhough, the improvement of VerifierQ over PRM looks still very marginal. The paper mentioned that this might be because of the small model size of Majority Vote. So, what is the model size for the other methods? Why the model size matters on this MATH dataset? It's not clear to me. For overestimation analysis, the paper said "Figure 6 reveals that PRM generally assigns higher Q-values to incorrect tokens, while VerifierQ assigns lower values", however, it feels to me that CQL in VerifierQ assigned low Q-values to all tokens where incorrect tokens got more penalization (lower Q-values) than the correct one. The ablation study should include the comparison of VerifierQ w/ CQL and VerifierQ wo/ CQL, in order to further show the mitigation in addressing overestimation. Overall, the conclusion is vauge.

**Questions:**

- See the comments in the weaknesses.
- How to get the "two tokens + and -" to represent correct and incorrect states, and the other "tag token indicating estimation"?
- In Equation 2, what's the difference between R(s,a) and r? How does equation 2 bound the Q-values within the range of (0,1)?
- What's the data for plotting Figure 3?
- What's the size of the action space for each dataset? Does the solutions per problem represent the action space for RL?
- "test accuracy %" should not include "%", for example, the numbers reported in Figure 4 should be the accuracy number in decimal instead of in percentage, right?

---

> ### Author Response · Authors · 2024-11-22
> **Partial Response 1**
>
> # Response to Reviewer Comments
>
> We sincerely thank you for your time and detailed review of our paper. We are glad that you find the focus on the verifier component interesting, and we believe that investigating the value model and temporal difference aspects can contribute significantly to the field. We have addressed your concerns point by point below.
>
> ## Weaknesses First Point:
>
> ### Presentation and Clarity:
> We actually put different modules and interactions in the updated Figure 2. The figure now illustrates how different modules interact within VerifierQ, highlighting both the forward and loss calculation for the backward passes.
>
> ### Utterance level MDP:
> We apologize for the lack of clarity regarding the utterance-level MDP. In the revised paper, we have added a detailed definition in Section 2 (Lines 88-97). Basically utterance level MDP would have States as the dialogue history of the sequence of utterances generated so far. For example, problem, solution step 1, solution step 2, etc. The Action is the next utterance, it is a complete sentence (a reasoning step). Reward is whether it is correct at that utterance. Transition is the concatenation of state and action, for example we have current solution steps and action which is the next solution step, then the next state is the concatenation of current solution steps and the next solution step.
>
> ### Utterance vs Token level MDP
> The key differences between utterance-level and token-level MDPs are:
> - **Granularity:** In token-level MDPs, each action corresponds to a single token, leading to a large number of steps to complete a solution (e.g., 100 steps for a 100-token solution). In contrast, the utterance-level MDP treats each complete sentence as an action, significantly reducing the number of steps (e.g., 5 steps for a solution with 5 sentences). In general, one sentence has 20-30 tokens.
> - **Action Space Size:** Token-level action spaces are equal to the vocabulary size, whereas utterance-level action spaces are exponentially larger due to the combinations of tokens forming sentences.
> - **Reward Frequency:** The correctness is usually determined at the end of a complete thought or sentence. Utterance-level MDPs align the reward signal more closely with actions, facilitating learning.

---

> > ### Author Response · Authors · 2024-11-22
> > **Partial Response 2**
> >
> > ## Weaknesses Second Point:
> >
> > ### Computational Efficiency
> > "I didn't find the results used for showing the computational efficiency of VerifierQ and others."
> >
> > We apologize for not providing explicit results on computational efficiency in the initial submission. In the revised manuscript, we have included a detailed computational efficiency analysis in Appendix C.1. In our preliminary experiments, we observed that VerifierQ significantly reduces training time compared to step-by-step Q-learning methods due to its ability to estimate multiple Q-values for multiple steps in a single forward pass. Specifically, in our preliminary experiments compared to the single step estimation we saw a big difference between training time after adapting to the multiple Q-value estimation. We include detailed analysis in the revised paper in Appendix C.1 to illustrate this efficiency gain, along with a discussion on how VerifierQ's architecture contributes to reduced computational overhead. Our initial observation aligns with the Big-O analysis.
> >
> > ### Performance Analysis
> > "In Figure 4, particularly on GSM8K dataset, VerifierQ performed worse than PRM when the number solutions per problem was less than 2^5, and it became the best when the number was larger than 2^5. However, the improvement looks very marginal."
> >
> > You are correct that VerifierQ performs slightly below PRM when the number of solutions per problem is less than 2^5=32. This is expected because VerifierQ is designed to leverage larger pools of generated solutions to evaluate and select higher-quality answers. It is similar to other works' comparison of PRM and majority vote. The difference gap increases with the number of solutions of increase. However in our case we show verifierQ and PRM's gap increase. As the number of solutions increases beyond 32, VerifierQ's performance surpasses PRM, demonstrating its strength in settings with more extensive test-time compute. The more generation budget the better and bigger the gap would be. Regarding the magnitude of improvement, we added a statistical significance test in section 6.2 and they are significant. VerifierQ and PRM results are here: GSM8K: $W = 1500.0, p < 1.64 × 10^{-35}$ MATH: $W = 156.5, p < 4.70 × 10^{-43}$. These results indicate that the improvements are statistically significant, even if they appear marginal in absolute terms. We have emphasized the importance of these gains in Section 6.2 of the revised paper.
> >
> > ### Model Size and Performance
> > "On MATH dataset, VerifierQ performed worse than Majority. Even though, the improvement of VerifierQ over PRM looks still very marginal. The paper mentioned that this might be because of the small model size of Majority Vote. So, what is the model size for the other methods? Why the model size matters on this MATH dataset? It's not clear to me."
> >
> > All methods evaluated, including VerifierQ, PRM, and Majority Voting, utilize the same underlying model: TinyLlama-1.1B. We chose this model due to computational and our resource constraints, ensuring a fair comparison of small size model across methods. Most other papers can run with minimum 7B and larger models. Our model size is 7 times smaller than others' smallest model size. The MATH dataset is notably more challenging than GSM8K, containing problems that require deeper reasoning and more complex calculations. Larger models have shown significantly better sample efficiency for learning as shown in [1]. We have clarified this point in section 6.2. So smaller model has inherent disadvantages given the same dataset due to lower sample efficiency.
> >
> > ### Overestimation Analysis
> > "For overestimation analysis, the paper said 'Figure 6 reveals that PRM generally assigns higher Q-values to incorrect tokens, while VerifierQ assigns lower values', however, it feels to me that CQL in VerifierQ assigned low Q-values to all tokens where incorrect tokens got more penalization (lower Q-values) than the correct one. The ablation study should include the comparison of VerifierQ w/ CQL and VerifierQ w/o CQL, in order to further show the mitigation in addressing overestimation."
> >
> > Thank you for this insightful observation. We believe CQL penalizes out-of-distribution actions, but it is particularly effective at penalizing incorrect tokens more than correct ones, thus improving the model's discriminative ability. We see the correct token is about the same value as PRM but the incorrect ones are lower. In addition, the gap between correct and incorrect ones is much larger.
> >
> > Your suggestion is very helpful, we added verifierQ without CQL term to the ablation study (VerifierQ(Base)) and also Appendix C.2. We can see from the gap between correct and incorrect almost doubled -- from around 12% to 30% of mean over estimated values. We move the relevant discussion to Appendix C.2 due to space limit.

---

> > > ### Author Response · Authors · 2024-11-22
> > > **Partial Response 3**
> > >
> > > ## Responses to Questions:
> > >
> > > ### Token Implementation
> > > How to get the "two tokens + and -" to represent correct and incorrect states, and the other "tag token indicating estimation"?
> > >
> > > We leave the details in Appendix A.1, and we added a bit more details in Appendix C.1. But the key idea is following the Math Shepherd paper [2]. LLM is predicting the next token, and we have the data labeled whether the step is correct or incorrect right after the tag token. So during the training, the model would predict what's the probability of + and - when tag token is present. During the inference, we extract the model's next token prediction after tag token from + and -.
> > >
> > > ### Equation Clarification
> > > In Equation 2, what's the difference between R(s,a) and r? How does equation 2 bound the Q-values within the range of (0,1)?
> > >
> > > We apologize for the confusion. In Equation 2, r(s,a) represents the immediate reward received after taking action a in state s. We have corrected the notation for clarity. We added bounding derivations in 5.1. Basically $0\leq r \leq 1$ and $ 0\leq Q \leq 1$, we have $0 \leq r + \gamma \max Q \leq 2$ for $\gamma \in [0,1]$. Thus we can bound  $0 \leq Q^*(s, a) = \frac{1}{2} (r(s,a) + \gamma \max_{a'} Q^*(s', a')) \leq 1$
> > >
> > > ### Figure 3 Data
> > > What's the data for plotting Figure 3?
> > >
> > > Figure 3 is an illustrative example created using synthetic data to demonstrate the differences between expectile regression and mean regression. It serves to provide an intuitive understanding of how expectile regression focuses more on the higher values, which is relevant to how IQL approximates the maximum Q-values without explicit maximization.
> > >
> > > ### Action Space Details
> > > What's the size of the action space for each dataset? Does the solutions per problem represent the action space for RL?
> > >
> > > The action space in our setting is extremely large due to the combinatorial nature of language:
> > >
> > > **Theoretical Action Space:**
> > > - At each step, the number of possible actions (utterances) is V^N, where V is the vocabulary size (e.g., 32,000 tokens) and N is the average length of an utterance (e.g., 20 tokens).
> > > - This results in an astronomical number of possible actions (e.g., 32000^20).
> > >
> > > **Practical Action Space:**
> > > - Realistically, the action space is constrained by grammatical and semantic validity, but it remains vast.
> > > - Solutions per problem represent a small subset of possible actions and are used as observed actions in the offline RL setting.
> > > - Many potential actions are unobserved, making it essential to handle the large action space effectively. You can think of it as how many different ways to answer a problem.
> > >
> > > ### Figure Label Correction
> > > "test accuracy %" should not include "%", for example, the numbers reported in Figure 4 should be the accuracy number in decimal instead of in percentage, right?
> > >
> > > You are correct. We apologize for this oversight. We have corrected the labels in Figure 4 to accurately represent the data, ensuring consistency between the labels and the values presented.
> > >
> > > ## References
> > > [1] Scaling Laws for Neural Language Models https://arxiv.org/abs/2001.08361
> > >
> > > [2] Math-Shepherd: Verify and Reinforce LLMs Step-by-step without Human Annotations https://arxiv.org/abs/2312.08935

---

### Author Response · Authors · 2024-11-22

# Response to Reviewers

We sincerely thank all the reviewers for their thoughtful and constructive feedback. We are grateful for their recognition of the strengths of our work, particularly in integrating reinforcement learning (RL) techniques with large language models (LLMs) to enhance multi-step reasoning capabilities. We appreciate that most reviewers consider this work as a good contribution to the research community. We have carefully considered all comments and have made significant revisions to address the concerns raised.

## Summary of Revisions and Improvements:

### Improved Writing Quality:
- Corrected grammatical errors and replaced informal language with formal academic writing throughout the manuscript
- Ensured consistent notation and terminology

### Enhanced Explanations:
- Expanded Section 2 to clarify utterance level MDP
- Expanded Section 5.1 to provide a detailed explanation of the modified Bellman update and how it bounds Q-values within [0,1], including mathematical derivations
- Expanded Section 6.2 to add more explanation, statistical significance test, and analysis
- Revised Section 7.1 to give more explanations and added Appendix C.1 and clarified computation efficiency
- Expanded and Renamed Section 7.3 so component namings are more descriptive to indicate what is being removed or modified. Also added more analysis
- Moved Section 7.4 to Appendix C.2 and expanded it due to space limit

### Figures and Visual Aids:
- Updated all figures with increased font sizes for better readability
- Redesigned **Figure 2** to include:
  - (A) Bellman update
  - (B) IQL and CQL components
  - (C) Illustrate the interactions between modules
- Corrected labels in **Figures 4 and 5** to accurately represent the data (e.g., correcting "Test Accuracy %" to "Test Accuracy" when using decimal fractions)

We are grateful for the reviewers' constructive feedback and positive comments, which have helped us improve our work. We hope that the revised manuscript meets the expectations of the reviewers and the conference.

---

> ### Author Response · Authors · 2024-11-25
>
> We find that diff-pdf on the OpenReview website might be difficult to use. To make it easier for you to review the changes, we have manually highlighted the key differences in the revised document.
>
> As the rebuttal period closes on Nov 26 for reviewers, we kindly ask for any additional feedback or clarifications regarding the updated submission.
>
> Your input is highly valuable, and we are happy to address any remaining questions or concerns.
>
> Thank you for your time and effort!

---

### Meta-Review · Area_Chair_puKS · 2024-12-23

**Metareview:**

The paper proposes to use Offline RL methods to learn verifiers for language generation. The approach is timely and addresses an important topic, and it would merit publication if concerns were better addressed. However, reviewers did raise major concerns regarding clarity, related work, evaluation, and ablation. The scores were widely spread (3, 5, 5, 8), but I find the highest score to be insufficiently justified by the corresponding brief review. Moreover, while I agree with the content of Reviewer M9KV's positive comment regarding novelty, I draw the opposite conclusion: utterance-level formulations have been studied before, and even if not in the context of verifier models, this significantly limits novelty.

**Additional Comments On Reviewer Discussion:**

Most reviews were thorough and raised important concerns. The authors seem to have addressed these concerns in their feedback, however only one reviewer agreed to follow up, and was ultimately unconvinced. While the authors raise valid points, I find that many of the concerns remain, and encourage the authors to improve the manuscript toward future publication.

---

### Decision · Program_Chairs · 2025-01-22

Reject